# Horizontal transport on the continental shelf driven by periodic rotary wind stress

Nathan Paldor<sup>1</sup> and Lazar Friedland<sup>2</sup>

<sup>1</sup>Fredy and Nadine Herrmann Institute of Earth Sciences, Hebrew University of Jerusalem, Jerusalem, 91904 Israel

<sup>2</sup>Racah Institute of Physics, Hebrew University of Jerusalem, Jerusalem, 91904 Israel

Correspondence: Nathan Paldor (nathan.paldor@mail.huji.ac.il)

Abstract. Wind driven circulation on a linearly sloping continental shelf is studied by employing the Lagrangian equations of motion forced by periodic rotary wind stress. The analysis yields explicit approximate expressions for the water column trajectories in the longshore and cross-shore directions, and these expressions are verified by numerical integration of the governing nonlinear equations. The periodic rotary wind stress generates a steady longshore drift directed with land to its left when the wind rotates counterclockwise at sub-inertial frequencies and with land to its right in all other frequencies. Counterclockwise rotation of the wind at the local inertial frequency results in a strong resonance manifested in very fast longshore drift.

# 1 Introduction

The fundamental and succinct, *f*-plane, theory developed by V. W. Ekman in Ekman (1905) decomposes the transport (i.e. the vertically averaged horizontal velocity) at the ocean surface driven by uniform wind stress into a steady component directed at right angles relative to the overlying wind and inertial oscillations. The theory considers a layer of uniform depth (thickness) at the ocean surface forced by overlying uniform (in time and space) wind stress. This assumption breaks down over the continental shelf, where the bottom slopes nearly linearly with distance from the shore, so the vertical averaging includes a thinner layer near the coast. The nonuniformity of the depth (thickness) of the water column greatly modifies the original theory developed by Ekman, in which all coefficients are assumed constant in time and space.

Ekman's original theory of ocean transport by time-independent wind forcing was extended to cases in which the coefficients in the governing equations vary spatially. These cases include the latitudinal variation of the Coriolis frequency: Paldor and Friedland (2023b); Paldor (2024) and the linear slope of the continental shelf under steady wind forcing: Paldor (2025). In view of the primary role played by wind forcing in the dynamics and transport on the shelf Allen and Smith (1981); Lentz and Fewings (2012) it is important to also examine the ramifications of temporal changes in the wind forcing over the shelf, which is the focus of the present theory.


Starting in the 1970s, a series of studies have extended Ekman's theory in layers of uniform depth to periodic wind stress forcing Gonella (1972); Craig (1989); Orlić (2011). The theories are based on the decomposition of the periodic wind stress into clockwise (CW) and counterclockwise (CCW) rotating components, and these theories clearly show that the effect of the CCW component differs dramatically from that of the CW component. To demonstrate this difference, Orlić (2011) wrote the wind stress as:

$$\tau(t) = \tau^x(t) + i\tau^y(t) = Ae^{i\omega t} + Be^{-i\omega t},\tag{1}$$

where  $\tau^x$  and  $\tau^y$  are the wind stress components in the x and y directions, respectively, A and B are the amplitudes of the CCW and CW components, respectively, and  $\omega > 0$  is the wind stress frequency. In this notation, the explicit expressions for the counterclockwise (CCW) and clockwise (CW) components of the resulting ocean surface transport are (see Eq. 7 in Orlić, 2011):

$$U + iV = \frac{A}{i\rho(f+\omega)}e^{i\omega t} + \frac{B}{i\rho(f-\omega)}e^{-i\omega t},$$
(2)

where  $\rho$  is the uniform water density and f is the local Coriolis frequency (assumed positive as in the northern hemisphere). These expressions describe the particular solution of the inhomogeneous equation to which the inertial oscillations associated with solutions of the homogeneous  $2^{nd}$  order equation should be added to solve a particular initial value problem. The singularity of the solution's CW component is evident when the frequency of the CW wind forcing is equal to the Coriolis frequency.

The solution also implies that, a periodic CW wind stress induces a surface transport directed  $90^{\circ}$  to the left of the wind direction (in the northern hemisphere) when  $\omega > f$  in which case the coefficient of  $Be^{-i\omega t}$  is  $\frac{i}{\rho(|\omega - f|)}$ . This occurrence of a current directed to the left of the wind forcing at some frequencies is completely missing from Ekman's original theory.

The present study focuses on basins with variable depth that are excluded from these results, so the results in a variable water thickness should be derived from solutions to the wind forced problem in basins with a sloping bottom. In addition to its effect on the oceanic response to periodic wind stress forcing, the sloping bottom also induces horizontal convergence (divergence), of the Ekman transport, which is balanced by the downwelling (upwelling) of water to (from) the deep ocean Paldor (2025). As is well known, the phenomenon of Ekman pumping that results from the curl of the wind stress in flat bottom basins (Gill, 1982; Vallis, 2017) has important implications for the physical properties (Cushman-Roisin and Beckers, 2011; Liu and Zhou, 2020; Almeida et al., 2021) and biogeochemical distribution (Vinayachandran et al., 2021) at the ocean surface and its communication with deeper layers. Clearly, the quantification of Ekman pumping is based on mass conservation in a fluid of uniform density (Gill, 1982; Cushman-Roisin and Beckers, 2011; Pedlosky, 2013; Vallis, 2017). The continuity equation can be combined with the solution of the horizontal transport to yield an expression for the pumping in terms of the curl of the wind stress divided by the Coriolis frequency (see, e.g., Equation 9.4.2 in Gill, 1982) which applies also to a uniform wind stress on the  $\beta$ -plane. These estimates originate from the Eulerian view of mass conservation in which the divergence of the horizontal velocity is related to the change in the height (volume) of a fixed mass of a fluid element.

Although the alternate Lagrangian framework adopted here provides a simple and intuitive form of the momentum equations, mass conservation in this framework is more complicated and less intuitive. The reason is that in this framework mass



conservation is based on the explicit expression of the coordinate transformation between the initial time and any subsequent time (Milne-Thomson, 1996; Bennett, 2006) and such explicit expressions are rarely available. The Lagrangian system of nonlinear momentum equations for a wind forced water column on the  $\beta$ -plane was greatly simplified recently by substituting the pseudo angular momentum for the zonal velocity (Paldor and Friedland, 2023a; Paldor, 2024), which yielded the required explicit expressions of the horizontal trajectory of a single column. With these explicit expressions of the coordinate transformation, mass conservation could be applied to estimate the sign and magnitude of the horizontal divergence. This view of mass conservation was successfully applied in other wind driven problem Paldor (2024, 2025) so the development of explicit analytical expressions of the coordinate transformation in the present study can presumably be also employed to calculate the time-dependent upwelling when the wind stress rotates (in space) periodically (in time).

This paper addresses the wind driven dynamics on the continental shelf when the wind forcing is periodically rotating in time while its amplitude is held constant. It is organized as follows: Section 2 presents the non-dimensional model Lagrangian equations, and the analysis of the dynamical equations. In Section 3, we solve the equations numerically, thus verifying the validity of the analytic results. The paper ends in Section 4 with a summary and discussion of the derived findings.

### 2 Model equations and analytic considerations

# 2.1 The Lagrangian single column model

The equations describing the changes in the horizontal velocity at depth z subject to a z-dependent viscous force and a uniform Coriolis frequency (known as the f-plane model since  $f = 2\Omega \sin(\phi)$ , where  $\Omega$  is Earth's frequency of rotation, is assumed constant, determined by setting  $\phi$  equal to  $\phi_0$  - a central latitude) are given by (see e.g. Gill, 1982; Vallis, 2017):

$$\frac{dx}{dt} = u$$
,  $\frac{dy}{dt} = v$ ,  $\frac{du}{dt} = f_0 v + \frac{1}{\rho} \frac{d\tau^x(z)}{dz}$ ,  $\frac{dv}{dt} = -f_0 u + \frac{1}{\rho} \frac{d\tau^y(z)}{dz}$ , (3)

where u and v are the components of the horizontal velocity vector,  $\underline{V}$ , in the x and y directions, respectively,  $\rho$  is the water density,  $f_0 = 2\Omega \sin(\phi_0)$  is the constant Coriolis frequency and  $\tau^x(z)$  and  $\tau^y(z)$  are the viscous stress forces in the x and y directions, respectively at depth z (Vallis, 2017; Cushman-Roisin and Beckers, 2011).

The z-dependence can be eliminated by integrating the equations between a lower boundary, z=-H, and the surface, z=0, which upon division by the layer thickness, H, yields the equations for the vertically averaged velocity  $(=\frac{1}{H}\int_{H}^{0}\underline{V}dz)$ . Clearly, in these vertically averaged equations, the stress terms appear only at z=0 and z=-H. At z=0 the stress is set to the stress applied by the overlying winds and at z=-H it is set to zero for large H (where the velocity is assumed to vanish) or to an assumed bottom friction for small H that reaches the bottom of the shallow basin.

We now set the x and y directions parallel and perpendicular to the shoreline, respectively, as in Fig. 1. The vertically averaged counterpart of system (3) in these directions is:

$$\frac{dx}{dt} = U, \quad \frac{dy}{dt} = V, \quad \frac{dU}{dt} = f_0 V + \frac{\tau^x(0) - \tau^x(-H)}{\rho H}, \quad \frac{dV}{dt} = -f_0 U + \frac{\tau^y(0) - \tau^y(-H)}{\rho H}, \tag{4}$$

where U and V are the vertically averaged velocities in the x and y directions, respectively.

On a linearly sloping continental shelf, such as that sketched in Fig. 1, the layer thickness is given by H(y) = Sy so the second terms on the RHS of the dU/dt and dV/dt equations in system (4) are:

$$\frac{\tau(0) - \tau(-Sy)}{\rho Sy},\tag{5}$$

where  $\tau$  denotes  $\tau^x$  or  $\tau^y$ . The shoreline, y=0, is a special case that differs fundamentally from all y>0 points, since at y=0 both the denominator and the numerator in Eq. (5) vanish. In contrast, at y>0 the denominator is finite and the bottom stress,  $\tau(-Sy)$ , can be neglected compared to the wind stress,  $\tau(0)$ , since away from the shoreline the bottom velocity is small. In the remainder of this work, we focus on the range y>0, which eliminates the "shoreline singularity" at y=0. The immediate vicinity of the shoreline that is affected by this singularity is shaded gray in Fig. 1 and, as will be shown below, the extent of this excluded range has a negligible effect on the solution in the rest of the shelf. The gray region near the shore is the terminus of the landward directed bottom flow that balances the seaward directed surface flow that originates from the forcing by the overlying wind stress.

Figure 1. The linearly sloping shelf, the wind stress vectors,  $\tau^x$ , shown here at angles  $\theta=180^\circ$  (blue thick arrow) and  $\theta=270^\circ$  (green thick arrow) relative to the +x direction. The angle,  $\theta$ , increases in agreement with the counterclockwise rotation of the wind (curved arrow). The offshore directed surface current and the compensating onshore directed bottom current (dashed arrows) are shown in an upwelling mode, associated with a constant wind stress at  $\theta=180^\circ$ .



These considerations imply that at y > 0 the dynamical system of wind driven flow on the continental shelf is given by:

00 
$$\frac{dx}{dt} = U$$
,  $\frac{dy}{dt} = V$ ,  $\frac{dU}{dt} = f_0 V + \frac{\tau^x(0)}{\rho Sy}$ ,  $\frac{dV}{dt} = -f_0 U + \frac{\tau^y(0)}{\rho Sy}$ . (6)

In the model under study,  $\tau^x(0)$  and  $\tau^y(0)$  are periodic in time, while their amplitude remains constant. We therefore let  $\tau^x(0) = \Gamma\cos(\theta)$  and  $\tau^y(0) = \Gamma\sin(\theta)$  where  $\Gamma$  is the constant dimensional amplitude of the wind stress and  $\theta = \omega t$  is its direction relative to +x. Although the frequency,  $\omega$ , is by definition positive, here we attach to it a sign that indicates whether the direction of the wind relative to the +x direction,  $\theta$ , increases or decreases. Thus,  $\omega > 0$  implies counterclockwise rotation of the wind (denoted by CCW in Orlić, 2011), while  $\omega 

appear on the RHS of the equations in the system (7) - (10), x(0) = 0 can be set without loss of generality, so the only initial condition affecting the dynamics is y(0).

A general analytical scheme that can provide insight on the solutions of the nonlinear system of equations (7) – (10) and on the trajectory of a water column on both the f-plane and the  $\beta$ -plane forced by a wind stress was developed recently for constant H in Paldor and Friedland (2023a). In this scheme the system is analyzed by combining equations (8) and (10) to a single  $2^{nd}$  order equation:

$$\frac{d^2y}{dt^2} = -\frac{\partial\Phi(y)}{\partial y} = -(D+y) + \frac{\epsilon\sin(\omega t)}{y},\tag{11}$$

where  $\Phi(y)=\int \left(D+y-\frac{\epsilon\sin(\omega t)}{y}\right)dy=Dy+y^2/2-\epsilon\sin(\omega t)\ln(y)$  is a potential determined by the RHS of Eq. (10). In the following, we will assume that the wind stress contributions in Eqs. (9) and (11) are perturbations, i.e.,  $D\approx D(0), y\approx y(0)$  and  $\epsilon/y(0)\ll y(0)$ . Note that the last condition can be satisfied even when  $\epsilon$  is O(1), provided y(0)>3. According to Eq. (9) for  $\epsilon/y(0)\ll 1$ , D varies slowly with time, and hence the same is true for  $\Phi$ , which implies that the dynamics of the (V,y) subsystem is that of a quasi-particle in a slowly varying potential. The following analysis focuses on the (V,y,D) subsystem, Eqs. (9) and (11) for slowly varying D.

#### 2.2 The linear solutions

The initial conditions detailed above imply that inertial oscillations (i.e. in the absence of body forces) are filtered out from the dynamics since for  $\epsilon=0$  a water column remains in its initial location at all t when V(0)=0 and D(0)=-y(0). The first step of the analysis is to linearize y(t) and D(t) about their respective initial values y(0) and -y(0), that is, to substitute:  $y(t)=y(0)+\delta y(t)$  and  $D(t)=-y(0)+\delta D(t)$  in Eqs. (9) and (11) where  $\delta y(t)$  and  $\delta D(t)$  are  $O(\epsilon/y(0))$ . The resulting linear system is:

$$\frac{d^2\delta y}{dt^2} = -(\delta D + \delta y) + \frac{\epsilon \sin(\omega t)}{y(0)},\tag{12}$$

$$50 \quad \frac{d\delta D}{dt} = \frac{\epsilon \cos(\omega t)}{y(0)}.$$
 (13)

One can look for solutions of this system that have the form:

$$\delta y = a\sin(\omega t) + G(t) \tag{14}$$

$$\delta D = b\sin(\omega t) \tag{15}$$

where the constants a and b as well as the function G(t) need to be determined. Substituting this form of solution in Eqs. (12) and (13) yields the system:

$$\frac{d^2G}{dt^2} - \omega^2 a \sin(\omega t) = -(b\sin(\omega t) + a\sin(\omega t) + G) + \frac{\epsilon \sin(\omega t)}{y(0)},$$
(16)

$$b\omega\cos(\omega t) = \frac{\epsilon\cos(\omega t)}{y(0)}. (17)$$

The  $O(\sin(\omega t))$  terms in Eq. (16) yield:

$$-\omega^2 a = -(b+a) + \frac{\epsilon}{u(0)} \tag{18}$$

while the remaining terms yield:  $\frac{d^2G}{dt^2} = -G$  i.e. oscillations at the inertial frequency,  $\pm 1$ , that appear due to the presence of the wind forcing in the  $O(\epsilon)$  terms. The initial conditions imposed on G are: G(0) = 0 and  $dG(0)/dt = -a\omega$  to ensure that  $\delta y(0) = 0$  and  $d(\delta y(0))/dt = 0$ . The resulting solution of G(t) that satisfies these initial conditions is:  $G(t) = -\omega a \sin(t)$ , which implies that

$$\delta y(t) = a\sin(\omega t) - a\omega\sin(t). \tag{19}$$

The solution for b in Eq. (17) is  $b = \frac{\epsilon}{\omega u(0)}$  which, upon substitution in Eq. (18), yields:

$$a = -\frac{\epsilon}{\omega y(0)(1+\omega)}. (20)$$

Substituting the expressions of a, b and G(t) derived above in the general expressions of y(t) and D(t) yields the solutions:

$$y(t) = y(0) - \frac{\epsilon}{\omega y(0)(1+\omega)} \left[ \sin(\omega t) - \omega \sin t \right]$$
 (21)

$$D(t) = -y(0) + \frac{\epsilon}{\omega y(0)} \sin(\omega t) \tag{22}$$

The solution for y(t) has a strong resonance effect near  $\omega = -1$ . As can be expected, the time-dependent solution of the problem has two fundamental frequencies: The inertial frequency, -1, (that is,  $-f_0$  in dimensional units) and  $\omega$ , the frequency of the wind forcing. The corresponding solutions for x(t) can be derived by inserting Eqs. (21) and (22) in Eq. (7).

Having completed the analysis of the  $O(\epsilon/y(0))$  terms in Eqs. (9) and (11) we turn now to the analysis of the  $O(\epsilon^2/y(0)^2)$  terms, which can be better compared to the solutions obtained by numerical simulations.

## 175 2.3 Second order effects in the wind stress and the drift in x

The constant drift in x appears if D+y in Eq. (7) has a constant component. Linear solutions (21) and (22) do not produce such a component. Thus, we include  $O((\epsilon/y(0))^2)$  terms in Eqs. (9) and (11) to get:

$$\frac{d\delta D}{dt} = \frac{\epsilon \cos(\omega t)}{v(0)} - \frac{\epsilon \cos(\omega t)}{v(0)^2} \delta y,\tag{23}$$

and

$$\frac{d^2 \delta y}{dt^2} = -(\delta D + \delta y) + \frac{\epsilon \sin(\omega t)}{y(0)} - \frac{\epsilon \sin(\omega t)}{y(0)^2} \delta y, \tag{24}$$

where for  $\delta y$  we use the oscillatory linear solution (19). Then from Eq. (23) one concludes that D is purely oscillatory and, therefore, a zonal constant drift can only result from the  $2^{nd}$  order contribution in y. The time-average of this  $2^{nd}$  order contribution in (24) yields  $\left\langle \epsilon \frac{\sin(\omega t)}{y(0)^2} \delta y \right\rangle = \frac{\epsilon^2}{2\omega y(0)^3(1+\omega)}$ . This constant  $2^{nd}$  order component contributes to the following constant drift in x according to Eq. (7):

$$U_d = \frac{dx}{dt} = \frac{\epsilon^2}{2y(0)^3 \omega (1+\omega)}$$
. (25)

This drift is much stronger (resonant) near  $\omega=-1$  and changes sign at this frequency in accordance with the resonance in  $\delta y$  at this value. The existence of this constant,  $2^{nd}$  order, longshore drift is quite surprising in view of the periodically rotating wind stress forcing and the traditional (steady)  $90^{\circ}$  angle between the wind stress and the surface transport. The direction of the drift is positive except for  $-1 < \omega < 0$  in which case  $\omega(1+\omega)$  in the denominator is negative. In an entirely different Eulerian model in which the "shelf" is merely a transition zone between a finite-depth open ocean and a shallower, uniform-depth coastal region, Loder (1980) proposed a heuristic explanation for the long-shelf drift in terms of the Eulerian long-shore velocity gradient on both sides of the "shelf". Although this explanation does not apply to the purely Lagrangian model studied here, the emergence of a long-shore drift over a sloping bottom seems to be of general applicability, but the dependence of the drift's direction on the forcing frequency probably characterizes only the present model.

Numerical solutions of system (7) - (10) that verify the validity of the analytic approximations developed here are presented in Sect. 3.

## 3 Numerical confirmations



Four examples comparing the theory (white dashed curves) and simulations (blue curves) near  $\omega=+1$  and  $\omega=-1$  i.e.  $\omega=\pm 0.8$  and  $\omega=\pm 1.2$  are shown in Figs. 2–5. The initial conditions used in all simulations are x(0)=0, V(0)=0, D(0)=-y(0) and y(0)=4. The simulations were carried out using MATLAB routine ODE45, which is based on Runge-Kutta formula (4,5), with relative and absolute tolerances of  $10^{-9}$ . The figures show the accuracy of the approximate analytic expressions (blue thick solid curves) compared to the simulated solutions (white thin dashed curves) even when  $\epsilon=0.5$  is not very small (recall that for realistic oceanic values  $\epsilon<1$  and that the small parameter is  $\epsilon/y(0)$ ). For smaller values of  $\epsilon$  the approximate analytic expressions are even closer to the numerical solutions. The x(t) plots (lower panels) also demonstrate the validity of the approximate longshore drift given by Eq. (25) shown by the green straight lines in these panels. As anticipated analytically, the longshore drift is directed in the -x direction when  $-1 < \omega < 0$  and in the +x direction when  $\omega < -1$  and  $\omega > 0$ . Regarding the other variables of the dynamics, in addition to the direction of longshore drift, the only other appreciable difference between positive and negative forcing frequencies is in the  $\delta D(t)$  curves (results not shown).

# 4 Summary and Discussion

In this work, we developed a theory of surface transport on a linearly sloping continental shelf forced by periodically rotating wind stress. The perturbative analysis is based on the smallness of the nondimensional amplitude of the rotating wind forcing. The shoreline singularity where the shelf's mean depth, H, vanishes does not appreciably affect the solution far from the shoreline, since the solution there is independent of the seaward extent of the region containing the singular point. Analysis and numerical simulations show that although the wind forcing is rotary and periodic, there exists a longshore drift directed in the -x direction when  $-1 


Figure 2. A comparison between direct simulations (dashed white curves) and the explicit expressions developed in Sec. 2 for small  $\epsilon$  (solid thick blue curves) of: y (upper panel) and x (lower panel) for  $\omega = 1.2$  and  $\epsilon = 0.5$ . The green straight line in the lower panel shows the approximate expression for the drift given in Eq. (25).

The resonance expected to dominate the dynamics at  $\omega=-1$ , that is, for a CW rotation at the local inertial frequency is evident in the drift in x. which is about 4 times larger near  $\omega=-1$  (lower panels in Figs. 4 and 5) than near  $\omega=+1$  (lower panels in Figs. 2 and 3). The ratio between the drift in x for  $\omega$  near -1 and  $\omega$  near +1 increases drastically when the frequency approaches these values. For  $\epsilon=0.1$  (to ensure that the perturbation analysis is valid), the ratio between the drifts in x for  $\omega=-0.95$  and  $\omega=+0.95$  is O(50) (results not shown). The resonance at  $\omega=-1$  originates from the general solution of the homogeneous equation (mentioned only briefly in the above analysis, which focuses on the particular solution of the inhomogeneous equation) that describes inertial oscillations that rotate clockwise at the local Coriolis frequency,  $f_0$ , denoted as -1 in the notation used here.

As discussed in the Introduction, in the case of a uniform H periodic rotary wind forcing yields a range of  $\omega$  in which the surface transport is directed **left** of the wind stress. To show that this counterintuitive result exists also on the shelf one has to compare U + iV = dx/dt + idy/dt to  $\tau^x + i\tau^y = \cos(\omega t) + i\sin(\omega t)$  in the particular solution (i.e. after eliminating inertial

**Figure 3.** Same as Fig. 2 but for  $\omega = 0.8$ 

oscillations from the solution of U+iV). Substituting Eqs. (21) (without the  $\sin t$  term) and (22) in Eq. (7) and adding to the resulting equation i time the derivative of Eq. (21) yields:

$$U + iV = \frac{\epsilon}{y(0)} \left( \frac{-i}{1+\omega} \right) (\tau^x + i\tau^y). \tag{26}$$

The surface transport on the LHS of this expression is directed to the left of the wind stress for  $\omega < -1$ . This result extends the result originally derived by Orlić (2011) in a basin of uniform H to the continental shelf. As noted above, the CW notation used in Orlić (2011) where only positive  $\omega$  values were allowed, is represented in the present study by negative  $\omega$  values (and CCW by positive  $\omega$  values).

This last result is of primary observational importance as it predicts that along a coast dominated by the daily transition from sea breeze during daytime to land breeze during nighttime the direction of the longshore drift is determined by the sense of wind rotation (CW or CCW i.e.  $\omega < 0$  or  $\omega > 0$ ) and by the latitude (that determines the Coriolis frequency). The theory developed here should be applied to observations of the trajectories of drogued surface drifters on the shelf under known wind conditions. Our results should also be compared to simulations by Ocean General Circulation Models (OGCM). However, the

**Figure 4.** Same as Fig. 2 but for  $\omega = -1.2$ 



reader is reminded that our model deals with the vertically averaged velocity so care should be exercised when comparing our results to OGCM simulations where a surface boundary layer and an inviscid interior exist.

From a theoretical perspective, mass conservation should be applied to the solutions found here to assess whether or not periodic rotating wind stresses induce upwelling or downwelling. In the Lagrangian formulation employed here, x(t) and y(t) are dependent variables that quantify the temporal changes in the coordinates of a particular water column. Thus, dU/dx and dV/dy do not have the Eulerian meaning of components of horizontal divergence and only imply the co-variation of pairs of dependent variables. In the Lagrangian framework, where the only independent variable is time, conservation of mass in an incompressible fluid is determined by the Jacobian of the coordinate transformation between 0 and t: h(0) = h(t)J(x(t),y(t);x(0),y(0)) where h is the height of the water column and  $J(x(t),y(t);x(0),y(0)) = \left(\frac{dx(t)}{dx(0)}\frac{dy(t)}{dy(0)} - \frac{dx(t)}{dy(0)}\frac{dy(t)}{dx(0)}\right)$  is the Jacobian of the coordinate transformation from (x(0),y(0)) to (x(t),y(t)) (Bennett, 2006; Milne-Thomson, 1996; Paldor, 2025). In Eq. (25) x(0) is an integration constant, so  $\frac{dx(t)}{dx(0)} = 1$  while y(t) is independent of x(0) so  $\frac{dy(t)}{dx(0)} = 0$  i.e.  $\frac{h(0)}{h(t)} = \frac{dy(t)}{dy(0)}$ . In Eq. (21) y(0) appears in the expression of y(t) - y(0) so  $\frac{dy(t)}{dy(0)} \neq 1$  and, depending sensitively on the value of  $\omega$ ,  $\frac{h(0)}{h(t)}$  can be larger than 1 or smaller than 1 i.e., the periodic stress can generate mean upwelling or downwelling.

**Figure 5.** Same as Fig. 2 but for  $\omega = -0.8$ 

Code and data availability. No new data were created or analyzed and no new codes were developed in this theoretical work.

Author contributions. NP - Initiation of project, Writing, Editing; LF - Analysis, Simulations, Editing

Competing interests. The authors declare that they have no conflict of interest

Acknowledgements. The authors are happy to acknowledge that no funding was received for this research.

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
