# Peer review of "Horizontal transport on the continental shelf driven by periodic rotary wind stress"

_EGUsphere, 2025_

## Author Comment (AC1)

Detailed response to referee#1's comments on: OS-2025-5187 "Horizontal transport on the continental shelf driven by periodic rotary wind stress" by Paldor and Friedland

The Referee comments are written below in black and the authors response in blue

Well written. It would be good to note that the change in direction of forced wind components at the inertial frequency was observationally verified in Weller (1981)(JGR, vol 86 C3 pages 1969-1977). A suggestion is to make it clear perhaps in lined 10-15 that the fluid is not stratified. Perhaps the abstract should include words noting northern hemisphere and homogenous fluid. For a coastal oceanographer the normal thinking might be of a surface wind-driven layer overlaying and a bottom boundary layer and the merging of the two as the water shoals. Any idea how stratification would change the solutions? and would a bottom boundary layer have a rectified current as well?

We thank the referee for accolade that our paper is well written and for his constructive suggestions. Specifically:

The discussion of Weller (1981) now appears in L42-43

A note emphasizing that our model addresses a fluid of uniform density now appears in L76 and in Abstract (L1)

A more detailed explanation of the set-up of our problem was added in L102-108

---

## Author Comment (AC2)

Detailed response to referee#2's comments on: OS-2025-5187 "Horizontal transport on the continental shelf driven by periodic rotary wind stress" by Paldor and Friedland

The Referee comments are written below in black and the authors response in blue

I read the paper with great interest. Through a theoretical approach, the authors found a mean drift driven by a variable wind over a sloping continental shelf. I have the following questions for the authors' reference.

Thank you for the comment "I read the paper with great interest".

The authors assumed that the bottom stress of a wind-driven current can be neglected, stated in Lines 92-93. This should be justified carefully. Near the coast (but beyond the gray region shown in Figure 1), the convergence/divergence of alongshore wind-driven Ekman transport induces a sea-surface slope, which in turn generates a barotropic current. Near sea bed, this barotropic current induces the bottom shear stress. It is exactly the bottom Ekman transport that drives the compensating shoreward current shown in Figure 1. Hence, it is highly questionable to exclude the bottom shear stress. Certainly, the authors can state that this study only consider regions not very close to the coast, thus bottom shear is not that important. However, this requires the water depth being at least three times greater than the Ekman frictional thickness. In this case, the wind-driven Ekman current actually cannot feel the sea bed. Unfortunately, the authors treated the water depth H as the Ekman thickness (Eq. 5), thus the Ekman transport feels the topography even in deep water, which is not ture.

In the revised version we added an entire paragraph in L102-108 where we explain how $H$ serves both as descriptor of the bottom topography **and** the depth of the Ekman layer.

Other issues are as follows.

1. This paper actually considers the movement of centroid of water column, instead of the surface water that is often focused in Ekman dynamics. Please state it clearly.

We now explain (L82-84) that the transformation from system (3) to system (4) is only valid for the vertically averaged velocity in the water column.

2. The overall mathematics was unclear to me (and perhaps to most readers). Some key steps were missing. Some examples will be given.

We hope that the explanations we added in the revised better clarify the mathematical procedures.

3. I don't see the necessity of introducing the variable D (=U+y). Can you explain in which way it simplifies the mathematics or makes the physics more transparent ?

This fundamental step is now elaborated in the paragraph straddling P5-6 and in particular the newly added L127-129

4. Why can the solution of (12-13) be written as (14-15) ? Equation (16) was split into (18) and the equation in Line 160, why? It excludes the possibility that G(t) could be associated with sin(omega*t). I can see that the authors intentionally split the solution into an oscillatory (at frequency omega) term and an inertial (at frequency f) term, then verified that they can obtain such a solution that satisfies the equations. It is necessary to prove that the solution of the equations is unique.

We rewrote this subsection and hope that it is clearer now. Briefly, Eq. (13) is now solved by integrating it once w.r.t. time (which yields Eq. (14)). For Eq. (12) we assume the form given in (15) – a combination of a solution of the inhomogeneous equation ($a * \sin(\omega t)$) and a general solution of the associated homogeneous equation (G(t) that solves into $A * \sin(t)$). We then solve each of these by equating the coefficients of $\sin(\omega t)$ and $\sin(t)$ in (12).

5. The derivation of (25) is unclear.

In the revised version we expand the explanation in the paragraph preceding Eq. (24) (which was (25) in the previous version). We hope it is now clear that oscillatory terms average out to 0 so the only contribution can arise from the last term on the RHS of (23)

6. It is unclear how one can know that D is oscillatory based on (23). It seems to me the second term on R.H.S. has a non-zero periodic mean.

This results directly from Eq. (13). See the response to comment #4.

7. It is unclear how one can get the relationship in Line 183 based on (24). If delta_y uses the oscillatory solution in (19) and delta_D is also oscillatory, the time-average of all terms should be zero.

Except for the $\sin(\omega t) * \sin(\omega t)$ where the second $\sin(\omega t)$ comes from $\delta y$ in the last term on the RHS of (23). This product averages out to ½ after using the trigonometric identity $\sin^2(\omega t) = (1 - \cos(2x)/2)$.

8. The authors compared the numerical and theoretical solutions to the equations in Section 4. It is unsurprising that they are consistent. What's more reasonable is to compare the theoretical results with the simulation of a hydrodynamic model (either 2D or 3D).

A note was added in L267-270 in which we propose the Eulerian problem as a sequel.